# Introduction of Solid Foods in Preterm Infants and Its Impact on Growth in the First Year of Life—A Prospective Observational Study

**DOI:** 10.3390/nu16132077

**Published:** 2024-06-28

**Authors:** Margarita Thanhaeuser, Melanie Gsoellpointner, Margit Kornsteiner-Krenn, David Steyrl, Sophia Brandstetter, Bernd Jilma, Angelika Berger, Nadja Haiden

**Affiliations:** 1Department of Pediatrics and Adolescent Medicine, Comprehensive Center for Pediatrics, Medical University of Vienna, 1090 Vienna, Austria; margarita.thanhaeuser@meduniwien.ac.at (M.T.); margit.kornsteiner.krenn@gmail.com (M.K.-K.); sophia.brandstetter@meduniwien.ac.at (S.B.); angelika.berger@meduniwien.ac.at (A.B.); 2Department of Clinical Pharmacology, Medical University of Vienna, 1090 Vienna, Austria; melanie.gsoellpointner@meduniwien.ac.at (M.G.); bernd.jilma@meduniwien.ac.at (B.J.); 3Department of Cognition, Emotion, and Methods in Psychology, Faculty of Psychology, University of Vienna, 1010 Vienna, Austria; david.steyrl@univie.ac.at; 4Department of Neonatology, Kepler University Hospital, 4020 Linz, Austria

**Keywords:** preterm infants, solid foods, growth, necrotizing enterocolitis, bronchopulmonary dysplasia, intraventricular hemorrhage, machine learning

## Abstract

The aim of this study was to investigate whether age at introduction of solid foods in preterm infants influences growth in the first year of life. This was a prospective observational study in very low birth weight infants stratified to an early (<17 weeks corrected age) or a late (≥17 weeks corrected age) feeding group according to the individual timing of weaning. In total, 115 infants were assigned to the early group, and 82 were assigned to the late group. Mean birth weight and gestational age were comparable between groups (early: 926 g, 26 + 6 weeks; late: 881 g, 26 + 5 weeks). Mean age at weaning was 13.2 weeks corrected age in the early group and 20.4 weeks corrected age in the late group. At 12 months corrected age, anthropometric parameters showed no significant differences between groups (early vs. late, mean length 75.0 vs. 74.1 cm, weight 9.2 vs. 8.9 kg, head circumference 45.5 vs. 45.0 cm). A machine learning model showed no effect of age at weaning on length and length z-scores at 12 months corrected age. Infants with comorbidities had significantly lower anthropometric z-scores compared to infants without comorbidities. Therefore, regardless of growth considerations, we recommend weaning preterm infants according to their neurological abilities.

## 1. Introduction

From around 6 months of age, the nutritional needs of infants can no longer be met by human milk or formula [1]. Therefore, the European Society for Pediatric Gastroenterology, Hepatology, and Nutrition (ESPGHAN) recommends the introduction of complementary foods between the 17th and the 26th week of life in term infants, but evidence-based guidelines for the introduction of solid foods in preterm infants are not yet available [2]. Guidelines on the optimal composition of complementary foods for preterm infants are also lacking [2].

The increased nutritional requirements of preterm infants after birth and following their discharge from hospital differ significantly from those of full-term infants [3]. This suggests that special nutritional requirements may also be needed during weaning to achieve adequate growth in preterm infants.

Numerous observational studies on weaning in preterm infants have been published, showing a trend towards very early initiation of complementary feeding in these infants [4,5,6,7,8]. The mean age at weaning was earlier than the recommended age for term infants in almost all published observational studies, with the youngest infants starting earliest. However, the associated growth outcomes have shown considerable variability [4,5,6,7,8] and introduction of solid foods before 4 months of age might be associated with an increased risk of allergy [2].

Only three randomized controlled trials on the introduction of solid foods in preterm infants have been published, one of them more than 20 years ago and one in an emerging country [9,10]. Thus, both studies cannot be compared to current or Western standards. The third study was published recently and included only infants with stable growth [11]. Data on infants with comorbidities due to preterm birth are lacking.

The primary aim of this observational study was to investigate whether the timing of the introduction of solid foods in preterm infants has an impact on growth in the first year of life. Our goal was to encompass infants with significant perinatal conditions that could potentially impact their stable growth as described in the literature, such as necrotizing enterocolitis (NEC) [12] and chronic lung disease (bronchopulmonary dysplasia—BPD) [13] or intraventricular hemorrhage (IVH) due to negative influences on suck–swallow rhythms [14]. We aimed to closely monitor the introduction of solid foods in these infants and assess the advantages of introducing solid foods at different ages and with varying compositions of complementary foods.

## 2. Materials and Methods

This was a prospective observational study in preterm infants with a birth weight < 1500 g and a gestational age < 32 weeks. Infants were recruited between April 2016 and November 2021 in the neonatal outpatient clinic of a level IV neonatal care unit of the Medical University of Vienna, Comprehensive Center for Pediatrics, Austria, after informed consent was obtained from the parents. Written informed consent from one parent was sufficient due to low risk for participants. The study was approved by the ethics committee of the Medical University of Vienna (EK: 1273/2016, date of approval 16 April 2016) and registered at clinicaltrials.gov (NCT02936219, 18 October 2016).

At enrollment, infants were stratified according to their type of milk feedings (human milk, formula milk, or mixed feedings). Based on the specific time chosen by each parent for the introduction of solid foods, we categorized infants into two groups: an early complementary feeding group (starting solid foods <17th week of life corrected age) and a late complementary feeding group (starting solid foods ≥17th week of life corrected age).

### 2.1. Study Participants

All infants with a birth weight < 1500 g and a gestational age < 32 + 0 visiting the neonatal outpatient clinic of the Medical University of Vienna were eligible for the study. Infants with gastrointestinal diseases such as Hirschsprung’s disease, congenital heart disease, major congenital birth defects, or chromosomal aberrations were excluded from the trial.

### 2.2. Study Visits

Families of participating infants were invited to study visits together with regular visits at the neonatal outpatient clinic at term, 6 weeks, 12 weeks, 6 months, 9 months, and 12 months corrected age. For the study flow, please refer to Appendix A.

Anthropometry was assessed under standardized conditions at every visit using a baby scale (Seca 376, Seca Germany, Hamburg, Germany) in lying position for weight. Length was measured in lying position (Seca 210, Seca Germany, Hamburg, Germany). Head circumference measurements were performed with a tape measure.

Nutritional intake was estimated based on 24 h recalls at term and at 6 weeks corrected age. Furthermore, three-day dietary records (a self-reported logbook with a food record on three consecutive days, including one weekend day) and the introduction of the main food categories were queried at 3 months, 6 months, 9 months, and 12 months corrected age.

Data on comorbidities of infants were retrieved from medical charts.

### 2.3. Primary Outcome

With this observational study, we aimed to identify current feeding practices in preterm infants after starting solid foods. The primary outcome of this study was to examine whether age at introduction of solid foods had an influence on length at 12 months corrected age. To detect a difference in length of 5% between study groups, the inclusion of 152 infants was necessary.

### 2.4. Secondary Outcomes

Secondary outcomes included other anthropometric parameters and their corresponding z-scores. Influences of different comorbidities (NEC ≥ grade 2, BPD defined as oxygen demand ≥ 36 + 0, retinopathy of prematurity—ROP ≥ grade 2, culture-proven sepsis, IVH ≥ grade 2) on the age at introduction of solid foods were assessed.

### 2.5. Baseline Characteristics

Maternal and infant baseline characteristics, as well as data on neonatal morbidities, were collected from medical charts. Data on parental education were collected in the follow-up visits and divided into three groups according to the highest education level of either the father or mother of the infant (primary, secondary, or tertiary school).

### 2.6. Statistical Analysis and Machine Learning Model

In general, absolute and relative frequencies were calculated for ordinal and nominal data obtained on any date of measurement, respectively. For continuous variables, mean and standard deviations were calculated, and graphical descriptive analyses included growth curves and a dependency plot. Dependencies between siblings of multiple birth were not considered in the descriptive analysis. To detect differences between study groups, the Chi2-test or the Mann–Whitney U-test was applied. For basic statistical analyses, the program JASP version 0.18.3 was used.

For the regression analysis of the primary outcome length and length z-score at 12 months corrected age, a machine learning model was fitted. Furthermore, another model was fitted to detect the most influential parameters on the timepoint of starting solid foods to predict age at weaning. A total of five machine learning-based statistical analyses were performed using Python version 3.11 [15]. There were two to predict length at 12 months corrected age with a continuous and a categorical variant of the variable age at introduction of solid foods, and there were two to predict length z-score at 12 months corrected age, again with a continuous and a categorical variant of the variable age at introduction of solid foods. For prediction of length and length z-score at 12 months corrected age, the following confounding variables were included: gestational age, sex, length z-score at term, type of feeding at 6 weeks corrected age, age at introduction of solid foods, height of mother and father, BPD, and NEC.

The fifth model was performed to predict age when starting solid foods and included the following confounding variables: gestational age, sex, type of feeding at 6 weeks corrected age, BPD, NEC, IVH, maternal country of birth, maternal age, and highest parental education.

Each analysis includes the following three parts: (1) prediction model training [16,17], (2) generalizability testing [18,19], (3) model analysis [20,21]. Exact details on the analyses can be found in Appendix B.

## 3. Results

### 3.1. Screening and Participants

During the 5.5 year study period between April 2016 and November 2021, 580 infants were screened. A total of 529 infants met the inclusion criteria; in 308 cases, the parents refused participation, and 3 infants had already started solid foods at their first appointment in the neonatal outpatient clinic. After 21 dropouts due to various reasons (i.e., withdrawal of consent, screening failure, lost to follow up, no data on starting solid foods or on the primary outcome), the final cohort consisted of 197 infants. According to their corrected age when starting solid foods, 115 infants were assigned to the early group and 82 to the late group (Figure 1).

### 3.2. Baseline Characteristics and Neonatal Morbidity

Table 1 shows the maternal and infant baseline characteristics, as well as data on neonatal morbidities.

Mothers of infants in the late feeding group had a significantly higher incidence of premature rupture of membranes (*p* = 0.049) and gestational diabetes (*p* = 0.038).

Infants in the early group started solid foods at a mean age of 13.2 weeks corrected age (CI 95% 12.7–13.8, ±3), while infants in the late group started at 20.4 weeks corrected age (CI 95% 19.8–21.0, ±2.9).

The percentage of male infants was significantly higher in the early feeding group (*p* = 0.026), whereas BPD rate was significantly higher in the late feeding group (*p* = 0.005). With a mean gestational age of 26 + 6 (CI 95% 26 + 3–27 + 2, ±2 + 0) compared to 26 + 5 (CI 95% 26 + 2–27 + 2, ±2 + 2), infants of the early feeding group were of similar gestational age but had a slightly higher birth weight (early group: mean 926 g, CI 95% 879–973, ±254, late group: mean 881 g, CI 95% 823–938, ±262, *p* = n.s.). Other parameters were comparable between study groups.

### 3.3. Primary Outcome

At 12 months corrected age, infants in the early group had a mean length of 75.0 cm (CI 95% 74.4–75.5, ±3.1); infants in the late group were 0.9 cm shorter with a mean length of 74.1 cm (CI 95% 73.4–74.8, ±3.3) (Appendix A). This difference was insignificant (*p* = 0.053). Figure 2A depicts length z-scores within the first year of life.

### 3.4. Secondary Outcomes

Appendix A, shows all anthropometric parameters assessed in the first year of life of infants, including corresponding z-scores. Figure 2B,C depict weight and head circumference within the first year of life. At 6 weeks corrected age, the late feeding group exhibited significantly lower z-scores for both weight and length compared to the early feeding group. Specifically, the weight z-score mean for the early group was −0.64 (CI 95% −0.85–−0.42, ±1.15), while for the late group, it was −0.96 (CI 95% −1.20–−0.71, ±1.08) (*p* = 0.043). For length, the early group had a mean z-score of −0.597 (CI 95% −0.85–−0.34, ±1.34), and the late group had −1.02 (CI 95% −1.34–−0.71, ±1.39) (*p* = 0.036). However, these differences occurred before the introduction of solid foods in both groups. At 3 and 6 months corrected age, length of infants was higher in the early feeding group (3 months corrected age—early group: mean 59.8 cm (CI 95% 59.3–60.4, ±2.8), late group: mean 59.1 cm (CI 95% 58.4–59.8, ±2.9), *p* = 0.041; 6 months corrected age—early group: mean 66.9 cm (CI 95% 66.4–67.5, ±2.9), late group: mean 65.8 cm (CI 95% 65.2–66.5, ±2.9), *p* = 0.008). However, this difference in length did not persist up to 12 months corrected age.

### 3.5. Influence of Comorbidities, Type of Feeding, and Birthweight on Introduction of Solids

Table 2 presents the corrected age in weeks at which solid foods were introduced to infants with various comorbidities. It was observed that among infants with different neonatal conditions, those diagnosed with BPD started weaning the latest, at a mean corrected age of 18.1 weeks. Additionally, the table illustrates variations in the age at weaning based on the type of feeding at 6 weeks corrected age and according to birth weight.

Overall, breastfeeding was more prevalent in the late group compared to the early group. Specifically, 28% of infants in the early group were exclusively breastfed, while 65% were formula-fed. In contrast, 34% of infants in the late group were exclusively breastfed, and 31% were on formula feeding. Additionally, infants who were exclusively breastfed began solid foods approximately two weeks later than those fed formula milk. Within our study cohort, the age at which weaning occurred was inversely related to birth weight, indicating that infants with lower birth weights started solid foods later.

Data on anthropometric z-scores in infants with different comorbidities are shown in Appendix A. Infants with NEC showed an especially good catch-up growth after starting solid foods.

### 3.6. Machine Learning Models

Table 3 shows the results of the four machine learning models used to predict the primary outcome length and length z-score at 12 months corrected age. Age at introduction of solid foods neither influenced length at 12 months corrected age nor its z-score. Also, there was no difference whether using the categorical variable (early vs. late feeding group) or the continuous variable (age at introduction of solid foods in weeks). Length at term, sex, and height of the mother showed a significant influence on the primary outcome length at 12 months of age, whereas the model fit was low for all four models.

The dependency plot in Appendix A shows the effect of age when starting solid foods on length at 12 months corrected age. Between 15 and 18 weeks corrected age, there is a turning point—starting on solid foods before has a slightly positive effect on length at 12 months corrected age; starting thereafter has a negative effect.

Another model for the prediction of age when starting solid foods was used (Appendix A), but other than BPD, no significant influential factors could be detected. Again, the model fit was very low.

## 4. Discussion

This study is a prospective observational analysis of preterm infants, exploring how the timing of introducing complementary foods affects length and other anthropometric measures during the first year of life. The introduction of solid foods had no impact on growth within the first year of life for infants with an average birth weight below 1000 g and a mean gestational age at birth of less than 28 weeks in both groups. However, at 3 and 6 months corrected age, infants in the early feeding group exhibited greater length, suggesting a temporary acceleration in length growth. Nevertheless, other anthropometric measurements showed no significant differences between the groups during the first year. Infants in both study groups showed a remarkable length and weight catch-up growth from birth until one year corrected age.

Our findings align with the majority of both interventional and observational studies on the introduction of solid foods in preterm infants and their impact on growth. Only a few studies have reported an effect of early introduction of solid foods on subsequent growth. An observational study by Brion et al. included infants with a gestational age < 28 weeks and compared infants receiving ready-made complementary foods with those receiving home-made complementary foods [22]. Infants were grouped according to the age at introduction of solid foods. For infants on ready-made complementary foods starting at <26 weeks corrected age, the z-scores for weight-for-length and BMI were highest at one year corrected age [22].

Another study that reported higher growth velocity is a randomized controlled trial by Marriott et al., published over 20 years ago [9]. In this study, infants were assigned to either a group following a preterm weaning strategy or a control group adhering to the current best practice at that time. Infants in the preterm weaning strategy group were started on solid foods at 13 weeks postnatal age and received complementary foods that were higher in energy density and protein content. Consequently, it is not surprising that these infants had a higher growth rate compared to those in the control group [9].

Additionally, two other interventional randomized controlled trials by Gupta et al. and Haiden et al. have explored early versus late introduction of complementary feeding in preterm infants [10,11]. Haiden et al. observed a transient faster weight gain in the early group, with a higher weight-for-age z-score at 6 months corrected age. However, both studies found no differences in anthropometric parameters, such as weight, length, or head circumference, at one year corrected age [11].

Many observational studies have also reported no impact of the timing of solid food introduction on growth [8,23,24]. Therefore, it can be concluded that the introduction of solid foods to preterm infants should not be solely based on growth considerations, a finding that is further supported by our results [11]. The timing of weaning should instead be determined by the infants’ neurological abilities and readiness.

Almost all observational studies indicate that preterm infants are being introduced to solid foods earlier than recommended for term infants, with the earliest weaning occurring in the most preterm-born infants. One study from the United States reported that preterm infants were more likely to be introduced to solid foods before 4 months corrected age [4]. Similar findings emerged from Australia, where the median time for starting solid foods was 14 weeks corrected age, whereas term infants were weaned 5 weeks later [5]. In a study from Italy, median age to start solids was 15 weeks corrected age, with 18% of infants weighing less than <5 kg at weaning [6]. An Austrian study by Hofstaetter et al. revealed that more than 50% of infants were on solid foods before 17 weeks corrected age, with 23% starting before 12 weeks [7].

In our cohort, the mean age of starting solid foods was 16.2 weeks, with almost 60% of infants starting before 17 weeks corrected age. The mean age of weaning was 13.2 weeks in the early group and 20.4 weeks in the late group. Infants in the late group had a lower birth weight and a lower z-score for weight and length at 6 weeks corrected age. Unlike the findings of the above-mentioned studies, birth weight in our cohort inversely correlated with age at weaning. We did not inquire about the reasons why parents chose to introduce solid foods early. This would have provided some insight. It is presumed that many followed their pediatrician’s advice, as was also noted in a study by Baldassarre et al. [25]. The authors highlighted significant variability in weaning advice from primary care pediatricians due to the absence of evidence-based guidelines [25].

To identify the factors that most influence the age at which solid foods are introduced, we used a machine learning model to predict age at weaning, incorporating a range of variables, including infant baseline characteristics, maternal factors, neonatal morbidity, and parental cultural and socioeconomic background. Despite the broad scope of factors considered, the model demonstrated poor predictive performance, suggesting key influencing factors were not included in our analysis. Notably, as mentioned above, we collected neither information on parents’ reasons for introducing solid foods nor data on the infants’ neurological abilities at the time of introducing complementary feeding, both of which could potentially have improved predictability. Among variables we did examine, only BPD showed a significant relationship with weaning age, indicating that infants with BPD were more likely to start solid foods later, possibly because of ongoing respiratory instability or problems in oromotor function after long-term respiratory support. Other factors such as sex, gestational age, type of nutrition at six weeks corrected age, IVH, NEC, parental educational levels, and maternal age or country of birth did not significantly influence age at introduction of solids foods in our cohort. Although we observed a significantly higher rate of breastfeeding at six weeks corrected age in the late feeding group, breastfeeding was not found to be an influential factor in predicting age at weaning.

### 4.1. Comorbidities

Percentages of infants with a birth weight < 1000 g were rather low in other observational studies on preterm infants and solid foods, except for the studies by Spiegler et al., Ribas et al., and Boscarino et al., which focused on VLBW infants, and Brion et al., which included only infants with a birthweight < 1000 g [8,22,23,26]. To the best of our knowledge, none of these studies specifically targeted infants with significant comorbidities related to preterm birth. In our study, the incidence of comorbidities was comparable between the two study groups, except for the rate of BPD. However, we observed a tendency for parents of infants diagnosed with NEC, BPD, and sepsis to introduce solid foods later than parents of infants without such comorbidities. Infants with NEC especially showed a good catch-up growth after starting solid foods.

### 4.2. Limitations and Strengths

In 2022, we published a randomized controlled trial focusing on the introduction of solid foods in preterm infants [11]. To ensure a uniform cohort, we excluded infants with conditions that could affect stable growth. Noticing the lack of research on post-discharge nutritional interventions in infants with comorbidities, we conducted this prospective observational study. A limitation of our study is the lack of data on the neurological readiness of infants for solid foods and the lack of information on the reasons behind parents’ decisions to introduce solid foods, whether due to the child’s interest, pediatrician recommendations, or other factors. This information would have significantly enriched our understanding.

To address the inherent limitations of our study’s design, we chose a powerful statistical approach using machine learning. This method focuses on prediction, i.e., out-of-sample inference using flexible, complex, high-dimensional models, that is both robust and credible rather than in-sample inference, as is typically found in statistical testing. We developed a prediction model for the primary outcome of length at 12 months corrected age and the corresponding z-scores, including variables such as the age at introduction of solid foods, nutrition at 6 weeks, sex, gestational age, length at term, height of mother and father, BPD, and NEC. However, the model fit was low, indicating that other unknown factors influencing length at 12 months corrected age were not captured.

Our study provides valuable data on the initiation of complementary feeding in a sizeable cohort of infants with a mean birth weight of <1000 g, including those with neonatal comorbidities. We have also assessed the nutritional data through monthly feeding protocols, although analyses of these findings are pending.

## 5. Conclusions

The timing of the introduction of solid foods did not affect growth in the first year of life in VLBW infants, regardless of the presence or absence of comorbidities. Preterm infants diagnosed with BPD and those who were breastfed at 6 weeks corrected age started solid foods the latest. Furthermore, there was an inverse relationship between birth weight and corrected age at the start of solid foods in our cohort.

Based on these findings, we conclude that the introduction of solid foods in preterm infants should be guided by the neurological abilities of the infants rather than their growth metrics or any neonatal morbidities.

## Figures and Tables

**Figure 1 nutrients-16-02077-f001:**
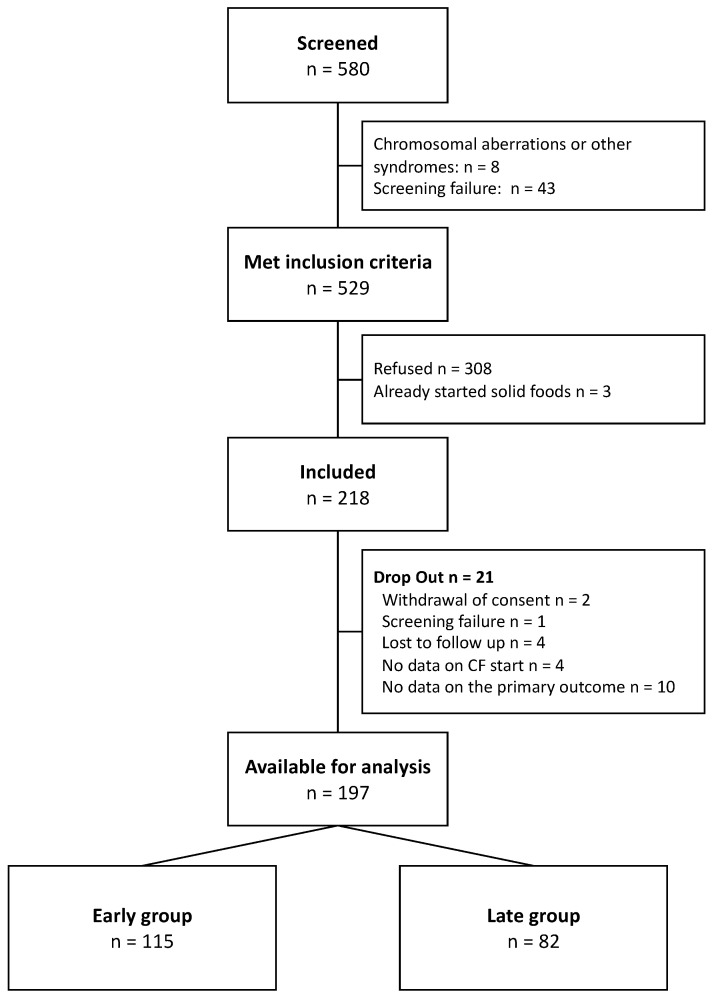
Patient flow chart.

**Figure 2 nutrients-16-02077-f002:**
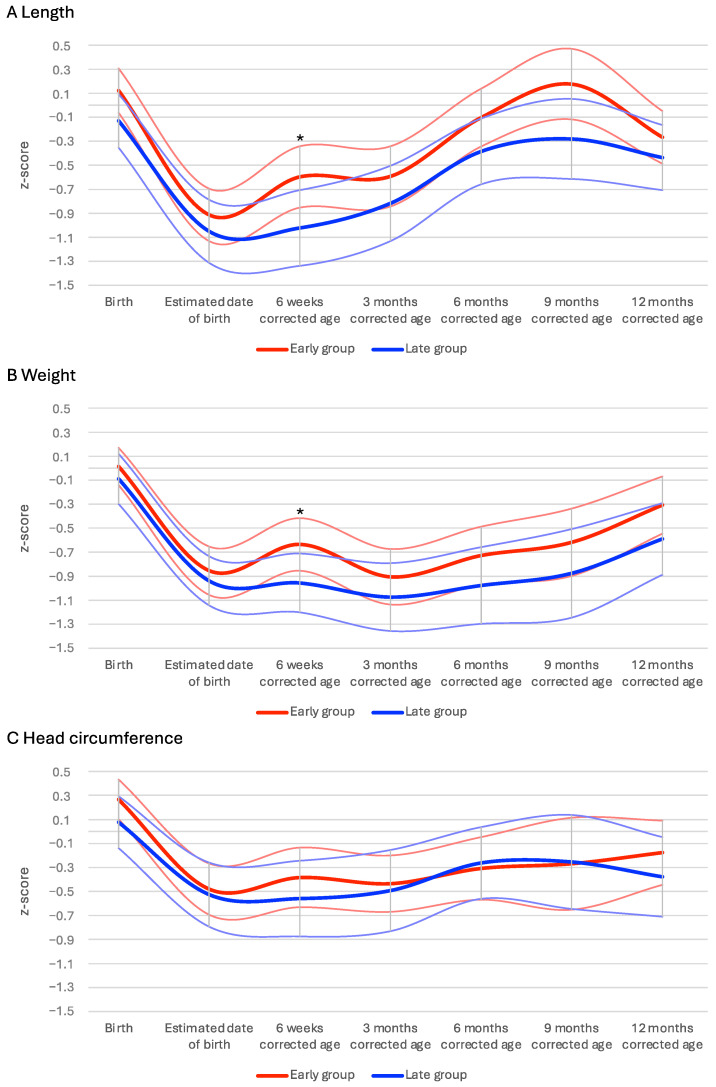
Z-scores of anthropometric data of the study population. Plot (**A**): this plot shows the length z-scores (mean and CI 95%) in the first year of life. The asterisks mark a significant *p*-value < 0.05. Plot (**B**): this plot shows the weight z-scores (mean and CI 95%) in the first year of life. The asterisks mark a significant *p*-value < 0.05. Plot (**C**): this plot shows the head circumference z-scores (mean and CI 95%) in the first year of life.

**Table 1 nutrients-16-02077-t001:** Baseline characteristics and neonatal morbidity.

Parameter	Early Group (n = 115)	Late Group (n = 82)
*Obstetric and parental parameters*		
Multiple pregnancy	36 (31.3%)	19 (23.2%)
Cesarean delivery	95 (82.6%)	70 (85.4%)
Prenatal steroids (any)	105 (91.3%)	76 (92.7%)
Premature rupture of membranes	43 (37.4%)	42 (51.2%) *
Gestational diabetes	0 (0%)	3 (3.7%) *
Preeclampsia	13 (11.3%)	17 (20.7%)
Age of mother at birth	31.4 (±5.8)	33.2 (±5.3)
Age of father at birth	35.1 (±7.2)	35.2 (±6.5)
Maternal education		
No graduation/school diploma	12 (10.4%)	8 (9.8%)
Middle school	32 (27.8%)	19 (23.2%)
Secondary school	23 (20%)	16 (19.5%)
Post-secondary school	43 (37.4%)	36 (43.9%)
Paternal education		
No graduation/school diploma	10 (8.7%)	8 (9.8%)
Middle school	45 (39.1%)	24 (29.3%)
Secondary school	21 (18.3%)	20 (24.4%)
Post-secondary school	33 (28.7%)	24 (29.3%)
*Neonatal parameters*		
Male sex	69 (60%)	36 (43.9%) *
Gestational age (days)	26 + 6 (±2 + 0)	26 + 5 (±2 + 2)
Birth weight (g)	926 (±254)	881 (±262)
Small for gestational age	4 (3.5%)	4 (4.9%)
*Neonatal morbidity*		
Necrotizing enterocolitis ≥ grade II	5 (4.3%)	6 (7.3%)
Bronchopulmonary dysplasia	14 (12.2%)	23 (28%) *
Persisting ductus arteriosus	51 (44.3%)	47 (57.3%)
Retinopathy of prematurity ≥ grade II	34 (29.6%)	27 (32.9%)
Sepsis, culture positive	16 (13.9%)	19 (23.2%)
Intraventricular hemorrhage ≥ grade II	17 (14.8%)	12 (14.6%)
Periventricular leukomalacia	0 (0%)	1 (1.2%)

Categorical data are presented as numbers with percentages in parentheses. Continuous data are presented as the mean and standard deviation in parentheses. * marks significant difference.

**Table 2 nutrients-16-02077-t002:** Weeks corrected age when starting solid foods in infants according to different comorbidities, type of feeding at 6 weeks corrected age, and birth weight.

		Total	Early Group	Late Group
**Morbidities**	NEC ≥ grade II (n = 11)	17.5 (±2.2)	15.8 (±1.6)	19 (±1.6)
	BPD (n = 37)	18.1 (±4.5)	14.1 (±3.2)	20.6 (±3.3)
	ROP ≥ grade II (n = 61)	16.9 (±4.1)	14.1 (±2.6)	20.4 (±2.7)
	Sepsis, culture positive (n = 35)	16.9 (±3.8)	13.6 (±2.3)	19.7 (±2.2)
	IVH ≥ grade II (n = 29)	16.9 (±4.2)	14.2 (±2.5)	20.8 (±2.9)
**Milk**	Breast milk (n = 62)	17.6 (±4.3)	13.8 (±1.7)	20.6 (±3.1)
	Mixed feedings (n = 33)	16.1 (±4.9)	12.4 (±3.1)	20.5 (±2.6)
	Formula (n = 97)	15.5 (±3.9)	13.5 (±3.1)	19.6 (±1.8)
**Weight**	<750 g (n = 62)	16.8 (±4.6)	13.7 (±3.2)	20.5 (±3.0)
	750–1000 g (n = 57)	16.0 (±4.0)	13.5 (±2.5)	20.1 (±2.3)
	>1000 g (n = 77)	15.9 (±5.0)	12.8 (±3.2)	20.5 (±3.2)

Continuous data are presented as the mean weeks corrected age with standard deviation in parentheses. BPD—bronchopulmonary dysplasia defined as oxygen demand > 36 + 0, IVH—intraventricular hemorrhage, NEC—necrotizing enterocolitis, ROP—retinopathy of prematurity.

**Table 3 nutrients-16-02077-t003:** Machine learning model, including influential factors for the prediction of length and length z-score at 12 months corrected age.

	Length at 12 Months Corrected Age				Length z-Score at 12 Months Corrected Age			
	Early group vs. Late group		Weeks corrected age at starting solids		Early group vs. Late group		Weeks corrected age at starting solids	
**Model fit**	R^2^ = 0.138		R^2^ = 0.134		R^2^ = 0.134		R^2^ = 0.125	
	*Effect size*	*p-value*	*Effect size*	*p-value*	*Effect size*	*p-value*	*Effect size*	*p-value*
Length z-score at term	1.03	**<0.001**	0.99	**<0.001**	0.39	**<0.001**	0.39	**<0.001**
Female sex	0.48	**0.001**	0.49	**<0.001**	0.09	0.116	0.09	0.157
Height of mother	0.3	**0.039**	0.26	**0.039**	0.11	**0.015**	0.1	0.066
Age at introduction of solids	0.19	0.181	0.14	0.560	0.04	0.542	0.03	0.843
Nutrition at 6 weeks	0.14	0.633	0.11	0.912	0.05	0.719	0.05	0.680
BPD	0.08	0.549	0.11	0.278	0.02	0.675	0.03	0.541
Height of father	0.06	0.939	0.09	0.922	0.03	0.917	0.04	0.942
Gestational age	0.05	0.885	0.07	0.858	0.02	0.921	0.03	0.916
NEC	0.03	0.915	0.01	0.894	0.01	0.905	0	0.866

For the columns “Early vs. Late group”, the variable “age at introduction of solid foods” was categorical with a cut-off at 17th weeks corrected age; for the columns “Weeks corrected age at starting”, the variable “age at introduction of solid foods” was continuous. Significant *p*-values < 0.5 are bold. BPD—bronchopulmonary disease with oxygen demand > 36 + 0 weeks corrected age, NEC—necrotizing enterocolitis.

## Data Availability

The study protocol and the individual participant data that underlie the results reported in this article, after de-identification, are available upon request from the corresponding author 6 months after publication. Researchers will need to state the aims of any analyses and provide a methodologically sound proposal. Proposals should be directed to nadja.haiden@meduniwien.ac.at. Data requestors will need to sign a data access agreement and, in keeping with patient consent for secondary use, obtain ethical approval for any new analyses due to ethical reasons.

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
