# Peer review of "Introduction of Solid Foods in Preterm Infants and Its Impact on Growth in the First Year of Life—A Prospective Observational Study"

_nutrients, 2024, doi:10.3390/nu16132077_

Round 1

Reviewer 1 Report

Comments and Suggestions for Authors

Thank you for allowing me to review this research, which is very interesting and even more so within the population group under study. The manuscript presents some aspects that should be better, but they are minor issues

Abstract

- There are acronyms that must be explained

Introduction

- Justify why these three pathologies and no others

- Assess the relationship between early introduction of food and allergic pathology

Methods

- Why less than 1500 grams? A 35 week will have a much lower percentile than a 28 week, then in the description it says <32 weeks, so at the beginning it must also be specified

- Why those exclusion criteria? Especially the topic of heart disease

- 2.5 has to go to results, not methods, it is data

Results

Ok

Discussion

Ok

Conclusion

- Always refer to preterm, not just children in general

Reviewer 2 Report

Comments and Suggestions for Authors

1)     The manuscript's title and abstract are specific and capture the content effectively.

2)     The abbreviations should be written in full the first time they appear in each section.   

3)     The authors should check the spelling and the manuscript's punctuation (For example, reference 18). The grammar of some sentences of the manuscript should be revised as well. (For example, in the first sentence of the abstract, the correct phrase should be "the aim of the study"), (In Table 1, the phrase "Education mother" needs to be revised).

4)     An expert should check the statistical analysis.

5)     In the discussion, the authors should explain a possible reason why neonates diagnosed with BPD were more likely to start solid foods later.

6)     The references are relevant to the study.

Comments on the Quality of English Language

Minor editing of English language required

Reviewer 3 Report

Comments and Suggestions for Authors

Dear Authors,
Congratulations on your effort to clarify if the age at which solid food is introduced influences the growth of very preterm infants. This prospective observational study brings more information on the effect of early solid foods introduction in the diet of extremely preterm infants with significant comorbidities, including NEC and BPD, conditions known to impact the growth and development of affected ELBW negatively. The use of five machine learning methods greatly supports the conclusions of this study. My only comment would be on breastfeeding. Please comment on the breastfeeding rates of the two study groups at the moment of solid food introduction. Could this be an important factor in delaying solid food introduction in ELBW and ELBW with significant comorbidities? Breastfeeding at 6  weeks seems a very distant moment when comparing the two groups.

Introduction - highlights the methods and important results of the study

Introduction - offers sufficient data for presenting the aim of the study

Material and Methods - are clearly described

Results - are offering a complete picture of the study

Discussions - the results of the study are discussed in the light of the results of previous data published on solid food introduction in preterm infants; limitations of the study are correctly presented

Conclusions are based on the study results

Supplementary material allows the screening of the complete evaluation of the two study groups.
